# Physics-Based Swab and Surge Simulations and the Machine Learning Modeling of Field Telemetry Swab Datasets

Amir Mohammad *, Mesfin Belayneh and Reggie Davidrajuh

Department of Electrical Engineering & Computer Science, University of Stavanger, 4021 Stavanger, Norway; mesfin.a.belayneh@uis.no (M.B.); reggie.davidrajuh@uis.no (R.D.)
* Correspondence: amir.mohammed@uis.no; Tel.: +47-951-29-664

**Abstract:** Drilling operations are the major cost factor for the oil industry. Appropriately designed operations are essential for successful drilling. Optimized drilling operations also enhance drilling performance and reduce drilling costs. This is achieved by increasing the bit life (minimizing premature bit wear), drilling more quickly, which reduces drilling time, and also reducing tripping operations. This paper is presented in two parts. The first part compares the parametric physics-based swab and surge simulation results obtained from the Bingham plastic, power law, and Robertson–Stiff models. The aim is to show how the model's predictions deviate from each other. Two 80:20 oil/water ratio (OWR) oil-based drilling fluids and two 90:10 OWR oil-based drilling fluids, 1.96 sg and 2.0 sg, were considered in vertical and deviated wells. Analysis of the simulation results revealed that the deviations depend on the drilling fluid's physical and rheological parameters as well as the well trajectory. Moreover, the model's predictions were inconsistent. Data-driven machine learning (ML) modeling is the focus of the second section. Data-driven modeling was performed using both software-generated datasets and field datasets. The results show that the random forest regressor (RF), artificial neural network (ANN), long short-term memory (LSTM), LightGBM, XGBoost, and multivariate regression models predicted the training and test datasets with higher R-squared and minimum mean square error values. Deploying the ML model in real-time applications and the planning phase would lead to potential applications of artificial intelligence for well planning and optimization processes.

**Keywords:** swab; surge; simulation; machine learning modeling; oil-based drilling fluids

## 1. Introduction

Poorly designed drilling operations can make drilling less efficient. This can cause problems such as damaged bits, slower drilling, twisted drill strings, and inaccurate measurement while drilling (MWD) tools. These issues can lead to unwanted round-tipping operations and drive up the cost of drilling. During tripping operations, the movement of the drill string in and out of the wellbore, making and breaking connections, results in undesired non-productive time (NPT) being spent, hence increasing drilling costs. It is logical to run the drill string up to its permissible threshold speed and shorten the time related to the tripping operation. However, drill string movement beyond the allowable speed will lead to well collapse and fluid influx to the wellbore while tripping out (the swabbing effect), and well fracturing while tripping in (the surging effect). The consequence of well collapse may cause drill string sticking. When the initial and alternative drill string unsticking operations are carried out without success, the final action is to locate the point of sticking, cut the drill string at the fee point, and then sidetrack. This results in a significant increase in the well budget.

Similarly, well fracturing results in drilling fluid losses. Here, the problem also increases the operational and non-productive time spent, increasing the overall drilling

budget. Predicting an appropriate well pressure mitigates possible well instability issues and kick influxes.

Over the years, researchers have built models to predict surge and swab effects based on different assumptions and conditions, such as steady-state and dynamic/transient conditions. Burkhardt (1961) developed a model to estimate surge and swab pressures for Bingham plastic fluids, considering steady-state conditions [1]. Schuh (1964) used a similar approach when developing a power law fluid model, assuming steady-state flow in a concentric annulus [2]. Fontenot and Clark (1974) developed a model to predict the swab pressure for Bingham plastic and power law fluids [3]. Mitchell (1988) produced a dynamic model that included several new factors, such as mud rheology, the elasticity of the pipe and the cement, the formation, changing temperatures, and viscous forces [4]. Ahmed et al. (2008) experimentally demonstrated the impact of pipe rotation on well pressure in an eccentric and concentric well filled with xanthan gum and polyanionic cellulose-based fluid [5]. Crespo et al. (2010) developed a simplified swab and surge model for yield–power law fluids [6]. Srivastav et al. (2012) experimentally showed that the speed of the trip, mud properties, annular clearance, and the eccentricity of the pipe highly affect the surge and swab pressures [7]. Gjerstad et al. (2013) employed a Kalman filter to predict and calibrate surge and swab pressures in real time for Herschel–Bulkley fluids based on differential pressure equations [8]. Ming et al. (2016) employed computational fluid dynamics techniques to build a swab and surge prediction model for concentric annuli. The comparison of simulations and experiments indicated an accuracy of up to 75% in predicting surge and swab pressures [9].

Fredy et al. (2012) utilized narrow slot geometry and regression techniques to develop a steady-state swab and surge prediction model, considering the compressibility of the fluid, formation, and pipe elasticity [10]. Erge et al. (2015) built a numerical annular pressure loss estimation model for eccentric annuli [11]. He at el. (2016) employed numerical simulations and regression techniques to forecast drilling operations' swab and surge pressures. The model indicated a $\pm 3\%$ maximum error compared with that of the experimental measurements [12]. Evren M. et al. (2018) utilized artificial neural network techniques and performed parametric studies on pressure loss [13]. Ettehadi et al. (2018) developed an analytical model for calculating pressure surges caused by drill string movement in Herschel–Bulkley fluids [14]. Shwetank et al. (2020) developed a two-layer neural network to predict swab and surge pressures [15]. Shwetank et al. (2020) also performed a parametric study to identify the impact of different parameters on the surge and swab pressures [16]. Zakarya et al. (2021) utilized numerical and random forest models to study the flow of drilling fluid through an eccentric annulus during tripping operations and the effect of eccentricity on annular velocity and apparent viscosity profiles [17]. Amir et al. (2022) employed deep learning techniques to predict the equivalent circulating mud density during tripping and drilling operations [18].

However, the reviewed scientific literature shows that the swab and surge models do not consider all the operational fluid properties and well geometry setups. Therefore, the applicability of swab surge models is valid for the considered assumptions and experimental setup conditions.

This study aimed to compare the predictions of swab and surge physics-based models, specifically the Bingham plastic, power law, and Robertson–Stiff models. The evaluation was conducted in vertical and deviated wells, using four different types of drilling fluids. Additionally, two machine learning models were applied to demonstrate how data-driven models can predict the synthetic physics-based dataset. We also implemented six machine learning (ML) models for the prediction of equivalent circulating mud density (ECD) in the actual field data acquired via a high-speed (wired drill pipe) telemetry system.

## 2. Swab Surge Modeling

This section concerns modeling. The first subsection is on physics-based modeling; the second subsection addresses machine-learning-based modeling.

### 2.1. Physics-Based Modeling

We used commercial software (DrillBench[TM] [19]) to evaluate the swab and surge behaviors in vertical and horizontal well profiles filled with different drilling fluids. Here, the experimental simulation setup is presented.

#### 2.1.1. Pore and Fracture Gradient

Figure 1 shows the pore and fracture pressure gradient window used for the swab and surge simulation. The experimental well was constructed with a 9 5/8″ casing set at 4000 m. For the swab and surge simulation study, the density of the drilling fluid was assumed to be at the mid-point between the tripping-in and tripping-out limits, as shown in the figure. The fracture gradient for the surging limit the casing was shown to be 2.06 specific gravity (sg), and the maximum pore pressure for the swabbing limit was 1.93 sg. For practical work, one needs to compute the collapse pressure gradient and compare it with the pore pressure to determine the swabbing limit. The collapse gradient was not calculated here because detailed information on the drilling formation was unavailable. Therefore, we assumed that the swab limit was based on the reservoir pressure. As shown in the figure, the fracture gradient increased as depth increased. However, this is not always the case. For instance, in the North Sea, there are weak formations, such as unconsolidated sandstone (e.g., Utsira sandstone), and tuff, which is volcanic ash (e.g., Balder formation). If these formations are present, the fracture gradient moves to the left side. Similarly, there are also mobile formations such as shale squeeze and salt formations. Hence, before commencing drilling operations, it is vital to have good knowledge of the geological formation to determine an appropriate well stability prognosis. Since there are many uncertainties that contribute to the well stability design, it is also important to include a safety margin for well collapse and fracture gradient curves. By doing so, one can predict the safe well pressure during static/circulation as well as during swab/surge operations. When the well stability window is narrow, the swab and surge speeds will be reduced. On the other hand, the wider window allows for higher swab/surge speeds, which reduce the NPT.

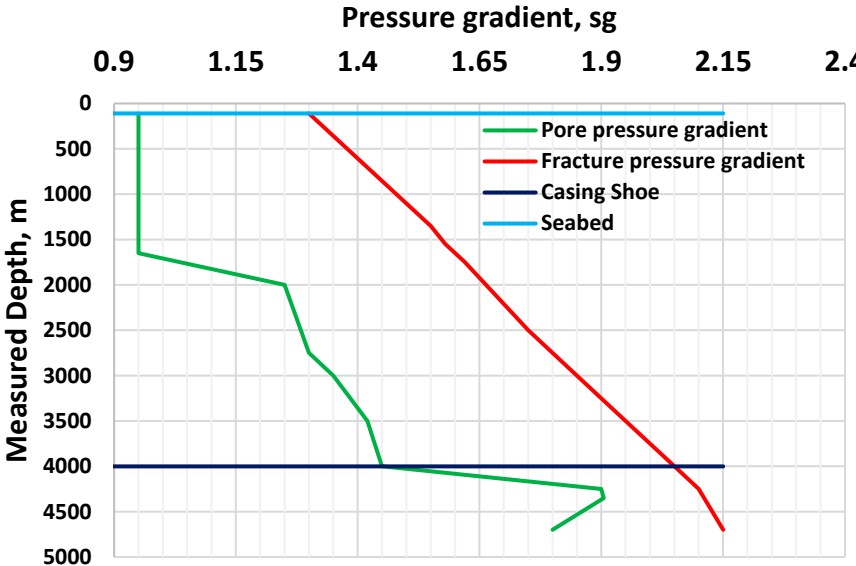

**Figure 1.** Pore and fracture pressure gradient prognosis.

#### 2.1.2. Experimental Well Construction and Well Trajectory

Figure 2 shows the 4800 m measured depth well construction design through which the swab and surge phenomena were simulated. The well was deviated with a maximum inclination of 36 deg with different azimuths. The survey data were obtained from a well drilled in Bangladesh [20]. The seabed was 109 m below sea level. A 21-inch riser was used to connect the surface with the seabed. A tapered drilling string that connected with 5″ at

the top and $3\frac{1}{2}''$ at the bottom was used. Drillbench$^{TM}$ software was used to construct the experimental well [19].

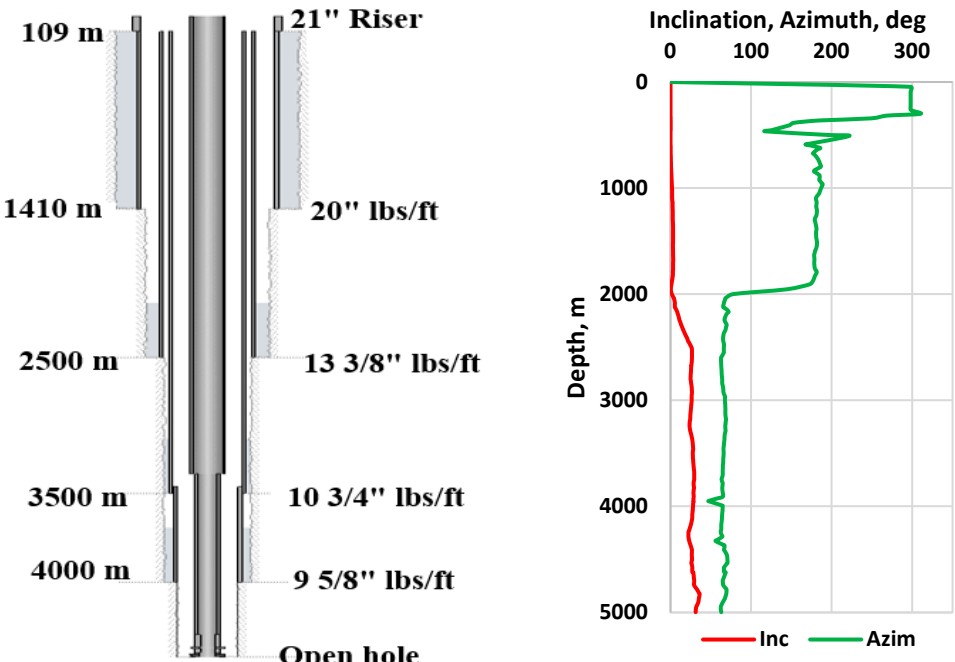

**Figure 2.** Experimental well structure and well trajectory.

### 2.1.3. Fluid Models

The swab and surge simulations were based on the hydraulic model. The model required defining PVT for density predictions and rheological models. Including tripping speeds, the density and viscosity parameters adjusted for the given temperature and pressure were used to calculate the pressure loss during swab and surge operations.

#### Fluid PVT Models

We used the empirical Glassø oil density model formulated based on North Sea oil for density calculation. For the water density model, we used the Dodson and Standing empirical model.

#### Rheology Models

Three rheological models were implemented in Drillbench$^{TM}$ software [19]. These were Bingham plastic (BP), power law (PL), and Robertson–Stiff (RS) models for the evaluation of swab and surge simulations.

Bingham plastic is a two-parameter rheology model. The model assumes a constant viscosity as the shear rate increases. The shear stress and shear rate vary linearly with excess yield stress. The model equation is as follows [21,22]:

$$\tau = PV\dot{\gamma} + YS \tag{1}$$

where PV is the plastic viscosity, $\dot{\gamma}$ is the shear rate, and YS is the yield stress.

The plastic viscosity and yield stress values were calculated from the 600 and 300 rpm dial readings of the Fann viscometer as:

$$PV[cP] = \theta_{600} - \theta_{300} \tag{2}$$

$$YS\left(lbf/100ft^2\right) = 2\theta_{300} - \theta_{600} \tag{3}$$

Unlike the Bingham plastic model, the shear stress–shear rate of the drilling fluid described by the power law excludes yield stress, and the viscosity decreases as the shear rate increases. The power law model is as follows [21,22]:

$$\tau = k\dot{\gamma}^n \tag{4}$$

where n and k are the flow and the consistency index parameters, respectively. The values were quantified based on curve fitting between the model and from the 600 and 300 RPM Fann dial readings as:

$$n = 3.321\log\left(\frac{\theta_{600}}{\theta_{300}}\right) \tag{5}$$

$$k\left(\text{lbf}/100\text{ft}^2\right) = \frac{\theta_{300}}{511^n} = \frac{\theta_{600}}{1022^n} \tag{6}$$

The Robertson–Stiff model is a three-parameter model. The model describes drilling fluids and cement slurries. The viscosity decreases as the shear rate increases [23].

$$\tau = A\left(\dot{\gamma} + C\right)^B \tag{7}$$

where A, B, and C are model parameters. *A* and *B* are similar to the *k* and *n* parameters of the power law model. However, the RS model exhibits yield stress, $\tau_o = AC^B$. C is the shear strain correction factor and can be determined using the interpolation method.

$$C = \frac{\left(\gamma_{min}\gamma_{max} - \gamma^{*2}\right)}{2\gamma^* - \gamma_{min}\gamma_{max}} \tag{8}$$

The shear rate, $\gamma^*$, can be calculated using the interpolation method, which is the corresponding value of $\tau^*$. The shear stress, $\tau^*$, is the geometric mean between the maximum and the minimum, given as:

$$\tau^* = \sqrt{\tau_{min}\tau_{max}} \tag{9}$$

### 2.1.4. Drilling Fluids

Figures 3 and 4 depict the four oil-based drilling fluids (OBMs) with different oil/water ratios (OWRs). Even though there were two pairs of drilling fluids with the same OWR (80:20 and 90:10), due to the different additive concentrations, all the fluids exhibited different viscosities. The fluids were used in the experimental well to assess the swab and surge model's prediction. To fit the well stability window shown in Figure 1, the density of the drilling fluids was elevated to 1.96 sg and 2.0 sg, maintaining the measured viscometer data. The fluids were filled in the vertical and the deviated experimental well. The 2.0 sg mud weights were approximately 0.06 sg and 0.07 sg away from the midline's lower (swab limit) and upper (surge limit) bounds, respectively. On the other hand, to simulate near overpressure, the mud weight was reduced to 1.96 sg, which is about 0.03 sg and 0.13 sg away from the lower and the upper limits, respectively.

The rheological parameters derived from the measurement and the rheology models were used as inputs for calculating the swab and surge hydraulics models. The three rheological models (Equations (1), (4), and (7)) were used to quantify the rheological parameters of the drilling fluids presented in Figures 3 and 4. Here, rheological modeling was based on curve fitting between the model and the measured viscometer dataset. The model prediction was evaluated by the absolute mean error percentile between the model calculated and the measurement as [24]:

$$Error_{ame} = \frac{1}{N}\sum_{i=1}^{N}\left|\frac{\tau_{measured} - \tau_{calculated}}{\tau_{measured}}\right| \times 100 \tag{10}$$

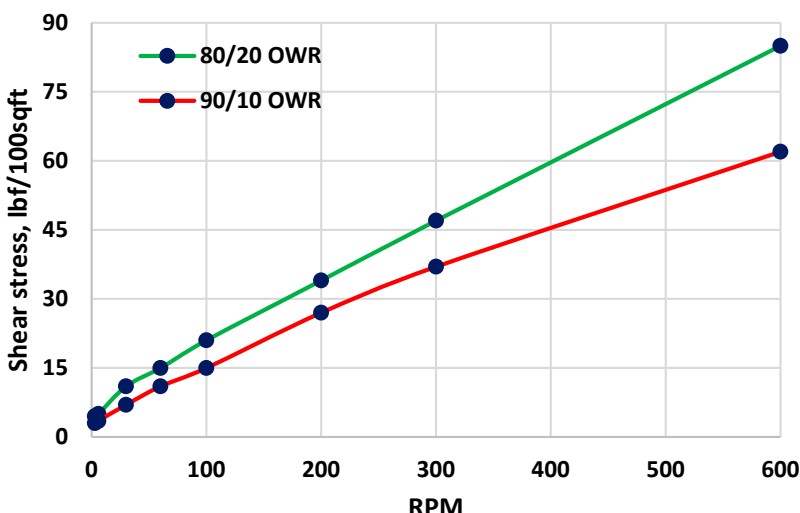

**Figure 3.** Fluid 1 viscometer data measured at 20 °C [25].

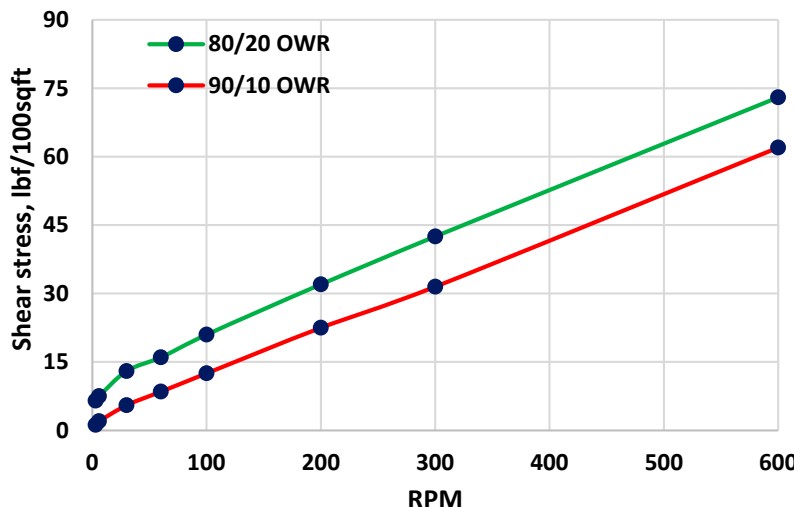

**Figure 4.** Fluid 2 viscometer data measured at 20 °C [26].

Tables 1 and 2 detail the rheological parameters and the percentile error derivations for fluid 1 (80:20 and 90:10, respectively) of the drilling fluids shown in Figure 3. The results show that the Robertson–Stiff model recorded a lower error deviation. This indicates that the model describes the drilling fluids better than the power law and the Bingham plastic models.

**Table 1.** The 80:20 OBM rheological parameters derived from Figure 3.

| Rheology Models | Parameters | | % Error |
|---|---|---|---|
| Bingham Plastic (BP) | YS [lbf/100sqft] | 6.061 | 13.2 |
| | PV [cP] | 40.314 | |
| Power Law (PL) | n [ ] | 0.546 | 13.6 |
| | k [lbfs$^n$/100sqft] | 1.656 | |
| Robertson–Stiff (RS) | A [lbfs$^n$/100sqft] | 0.262 | 1.5 |
| | B [ ] | 0.838 | |
| | C [s$^{-1}$] | 26.670 | |

**Table 2.** The 90:10 OBM rheological parameters derived from Figure 3.

| Rheology Models | Parameters | | % Error |
|---|---|---|---|
| Bingham Plastic (BP) | YS [lbf/100sqft] | 4.843 | 21.9 |
| | PV [cP] | 29.82 | |
| Power Law (PL) | n [ ] | 0.570 | 12.2 |
| | k [lbfs$^n$/100sqft] | 1.080 | |
| Robertson–Stiff (RS) | A [lbfs$^n$/100sqft] | 0.179 | 2.1 |
| | B [ ] | 0.855 | |
| | C [s$^{-1}$] | 24.418 | |

Similarly, Tables 3 and 4 show the rheological parameters and the percentile error deviations of the fluid 2 systems displayed in Figure 4, respectively. Again, the Robertson–Stiff model exhibited a lower error deviation.

**Table 3.** The 80:20 OBM rheological parameters derived from Figure 4.

| Rheology Models | Parameters | | % Error |
|---|---|---|---|
| Bingham Plastic (BP) | YS [lbf/100sqft] | 8.690 | 11.1 |
| | PV [cP] | 33.13 | |
| Power Law (PL) | n [ ] | 0.435 | 12.2 |
| | k [lbfs$^n$/100sqft] | 3.005 | |
| Robertson–Stiff (RS) | A [lbfs$^n$/100sqft] | 0.404 | 2.3 |
| | B [ ] | 0.751 | |
| | C [s$^{-1}$] | 40.48 | |

**Table 4.** The 90:10 OBM rheological parameters derived from Figure 4.

| Rheology Models | Parameters | | % Error |
|---|---|---|---|
| Bingham Plastic (BP) | YS [lbf/100sqft] | 1.724 | 13.6 |
| | PV [cP] | 30.26 | |
| Power Law (PL) | n [ ] | 0.720 | 7.7 |
| | k [lbfs$^n$/100sqft] | 0.384 | |
| Robertson–Stiff (RS) | A[lbfs$^n$/100sqft] | 0.138 | 5.6 |
| | B [ ] | 0.884 | |
| | C [s$^{-1}$] | 8.890 | |

However, the lower error deviation for the rheological model prediction does not mean that the hydraulic model accurately predicts the well pressure; Jeyhun et al. (2016) have previously demonstrated this [27].

### 2.2. Machine Learning Modeling

Figure 5 summarizes the modeling and performance evaluation process utilized in this paper. The first step was field data pre-processing to clean and select features for ML modeling. The second step was ML modeling, performed by splitting 70% of the dataset for training the model and the remaining 30% for testing the model. Seven ML algorithms were selected: polynomial, multivariate regression, LSTM, LightGBM, XGboost, artificial neural network, and random forest. Finally, the model performance accuracy was evaluated with a coefficient of determination ($R^2$) and mean square error. All the ML models and the model's accuracy performance analysis were simulated using Python Built-in Libraries. Therefore, only brief descriptions of both models, their concepts, and how they work are presented.

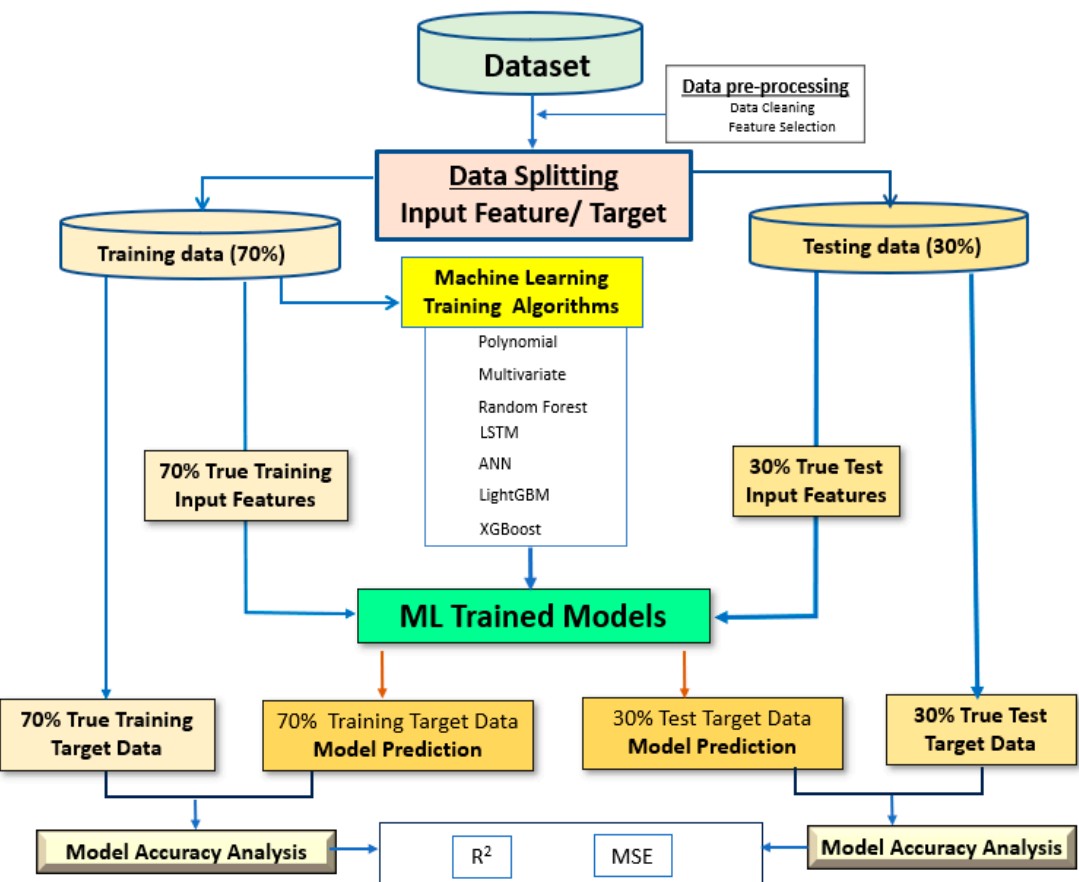

**Figure 5.** ML modeling and performance analysis workflow implemented in this study.

2.2.1. Polynomial Regression

The laboratory and the simulated swab and surge pressure variation with the tripping speed behave as a polynomial function. To correlate the single input feature (tripping speed) with the target output ($P_w$), the polynomial mapping function can be written as:

$$P_w = \beta_0 + \beta_1 V_p + \beta_2 V_p^2 \tag{11}$$

where $\beta_0$, $\beta_1$, and $\beta_2$ are the curve fitting parameters to be determined by the least sum square error method.

2.2.2. Multivariate Regression

For the field dataset, several features affect the ECD in the wellbore. Therefore, we used multivariable regression for multiple independent variables/features ($x_1$, $x_2$, $x_3$... $x_n$) to predict the target variable, y (ECD). The multiple linear regression model is the linear combination of the weighted features, and is written as (Anderson T.W., 2003) [28]:

$$y = \beta_0 + \beta_1 x_1 + \cdots + \beta_n x_n + \varepsilon \tag{12}$$

where y is the predicted value of the dependent variable, $\beta_0$ is the y-intercept (value of y when all other independent variables are set to 0), $\beta_1$ is the regression coefficient of the first independent variable $x_1$, $\beta_n$ is the regression coefficient of the last independent variable $x_n$, and $\varepsilon$ is the model error (how much variation there is in our estimate of y). The regression coefficients were determined by the least square error method.

### 2.2.3. Random Forest

Random forest regression is a supervised machine learning algorithm. Breiman (2001) outlined the concept behind the random forest algorithm: it builds decision trees based on different inputs and takes their majority vote for classification and averages in the case of regression [29]. As per Han et al. (2011), random forests reduce overfitting since they average over the independent trees [30]. In this study, the random forest regression model was applied for both field and synthetic data.

### 2.2.4. Artificial Neural Network

The ANN model was built using a feed-forward backpropagation network. The training algorithm used in this study was ReLu. In addition, the mean sum error square (MSE) loss function was employed to calculate the changing weight and update the weight change and a new learning state. The network was built with three layers: an input, a hidden layer, and an output layer. For the synthetic data, the input layer consisted of a single neuron (i.e., tripping speed (Vp)), the hidden layers comprised three neurons, and the output layer had one neuron (i.e., ECD). Multiple input layers were the running speed and bit position for field data; the output layer was ECD. An ANN model was also implemented for synthetic and field datasets.

### 2.2.5. LightGBM

LightGBM algorithm is a Python gradient-boosting decision tree framework. The gradient algorithm was selected to faster train data and for higher efficiency. Moreover, it exhibits lower memory usage, improved accuracy, and the ability to handle large-scale data [31]. LightGBM algorithm employed to the field dataset.

### 2.2.6. XGBoost

XGBoost is a family of gradient tree-boosting algorithms and an ensemble learning technique. The method was initially developed by Chen et al. (2016) [32]. Gradient boosting combines several weak models to obtain a robust model. Decision trees are trained in a series of iterations using weak XGBoost models. After each iteration, the method updates the model with a new decision tree to fix the flaws in prior trees. Each tree attempts to anticipate the residual of the proceeding trees as the trees are trained on those errors. One of the advantages of XGBoost is its capability to tackle missing data. It divides the data into left and right branches at each decision tree node and automatically learns how to handle any missing values [33]. XGBoost algorithm employed for the field dataset.

### 2.2.7. LSTM

The long short-term memory algorithm is formed of a recurrent neural network (RNN). It has a high capacity to capture underlying correlations among the different features in the data [34]. The most widely used LSTM algorithm is gradient-based backpropagation; it has a faster learning rate in real-time implementation [35]. We used LSTM ML modeling for the field dataset.

### *2.3. Model Accuracy Evaluation*

The model performance accuracy was evaluated with the commonly used statistical parameters of mean square error (MSE) and regression coefficient ($R^2$), as recommended by Montgomery (2019) [36].

### 2.3.1. Mean Square Error (MSE)

The mean square error (MSE) assesses the average squared difference between the observed and predicted values. When a model has no error, the MSE equals zero. As model

error increases, its value increases. The mean square error is also known as the mean square deviation (MSD), which is not explained by the regression model.

$$MSE = \frac{1}{N} \sum_{i=1}^{N} \left( y_i^{predicted} - y_i^{Actual} \right)^2 \tag{13}$$

Both ordinary linear regression and ANN ML algorithms use the loss, which is the sum of the difference between observed and predicted values to optimize the regression coefficients.

### 2.3.2. Regression Coefficient ($R^2$)

R-squared ($R^2$) is also known as the coefficient of determination. It defines the degree of variance in the dependent variable (output/target) that can be explained by the independent variable (input features).

$R^2$ values vary from 0 to 1. A score of 1 is the ideal, where 100% variation can be explained by the input feature variable. It provides the best goodness-of-fit line.

$R^2$ is the ratio of the sum of squares of residuals from the regression model (SSR) and the total sum of squares of errors from the average model (TSS), then subtracted from 1.

$$R^2 = 1 - \frac{\sum_{i=1}^{N} \left( y_i^{predicted} - y_i^{Actual} \right)^2}{\sum_{i=1}^{N} \left( y_{Actual}^{Mean} - y_i^{Actual} \right)^2} \tag{14}$$

### 2.4. Description of Tripping-Out Data

The dataset used in this study was obtained from a field located on the Norwegian shelf. The data were recorded by multiple downhole sensors located at different positions along the wellbore, as shown in Figure 6. The pressure sensors mounted on the enhanced measurement system (EMS) tool transferred the measured information to the surface via a wired drill pipe bidirectional telemetry system that transmits data at 57,000 bps (Reeves et al.) [37].

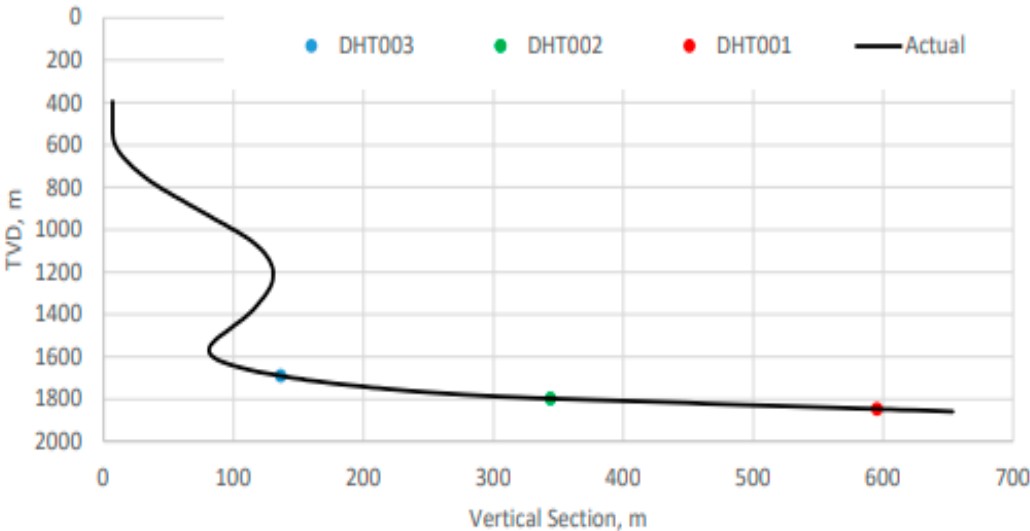

**Figure 6.** Sensor placement along the wellbore.

A Data While Tripping Tool (NOV DWT) was employed to assess the connectivity of the wired drill string throughout the tripping operations. The DWT made it possible to stay connected during the tripping operations, which has long been a desire of E&P companies to collect the downhole pressure and temperature data [38].

The trip was made from 1060 m to 330 m in an 8.5″ hole section. The downhole tool (DHT) sensors with the NOV company code names DHT001, DHT002, and DHT003 were

placed at 67 m, 328 m, and 575 m above the bit, respectively. In this study, the data used for modeling were only from sensors closer to the bit (i.e., DHT001). As shown in Figure 7, 3373 of the values consisted of the downhole equivalent mud weight (EMW), bit position, and running speed (VP).

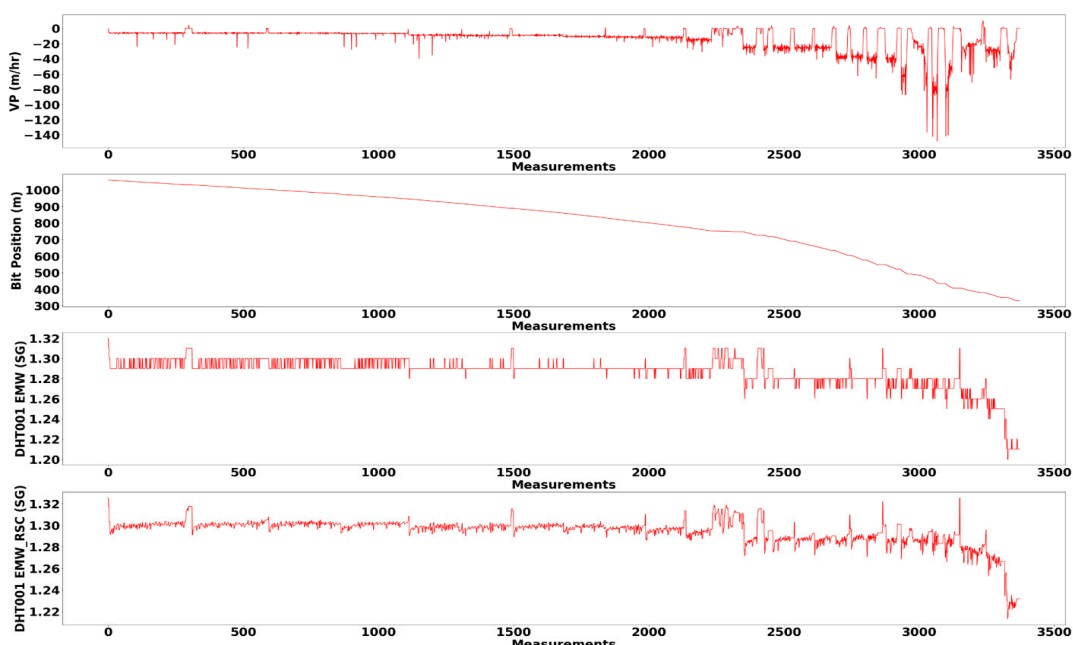

**Figure 7.** Tripping out dataset.

## 3. Results

This section presents the physics-based simulation and the machine-learning-based modeling results and analysis.

### 3.1. Physics-Based Simulation Results

#### 3.1.1. Result 1

The first computer swab and surge experiments were conducted using the drilling fluids shown in Figure 3. The simulations were performed at different tripping speeds while circulating. Figure 8a,b show the critical allowable swab and surge tripping speeds that led to the lower limit (i.e., at 1.93 sg) and the upper limit at the casing shoe (i.e., 2.06 sg), respectively. The presented examples were obtained from the results in deviated wells filled with 2.0 sg 80:20 and 90:10 OBMs. The comparison shows that the model prediction varied in the different fluid types. For instance, in the well filled with 80:20 OBM, the swabbing pressures obtained from Robertson, Stiff, and Bingham plastic were almost equal; however, the power law model underpredicted the results. For the 90:10 OBM, the power law and the Robertson swab speed predictions were closer, but the Bigham plastic deviated from the two models. On the other hand, surge speeds show different behaviors. Another observation is that the 80:20 OBM surging pressure decreased as the flow rate increased. In contrast, the three models in the 90:10 OBM showed an increase in surging pressure when the flow rate increased up to 300 lpm and then decreased as the flow rate increased. The simulation results reveal that one cannot conclude which hydraulic model to trust when designing the swab and surge model unless they are compared with measurement. The simulation results also showed that the swabbing speed was higher during circulation, and the surging speed was higher when there was no circulation. To reduce the tripping time, it was important to trip out while in circulation and trip in while de-activating the rig pump. For further model performance evaluation, the swabbing speed was selected when the circulation flow rate was 600 L per minute (lpm), and the surging speed was selected without the flow rate. The results are presented in Figures 9–12. The results

show the swabbing speed in the deviated and vertical wells filled with the two different drilling fluids.

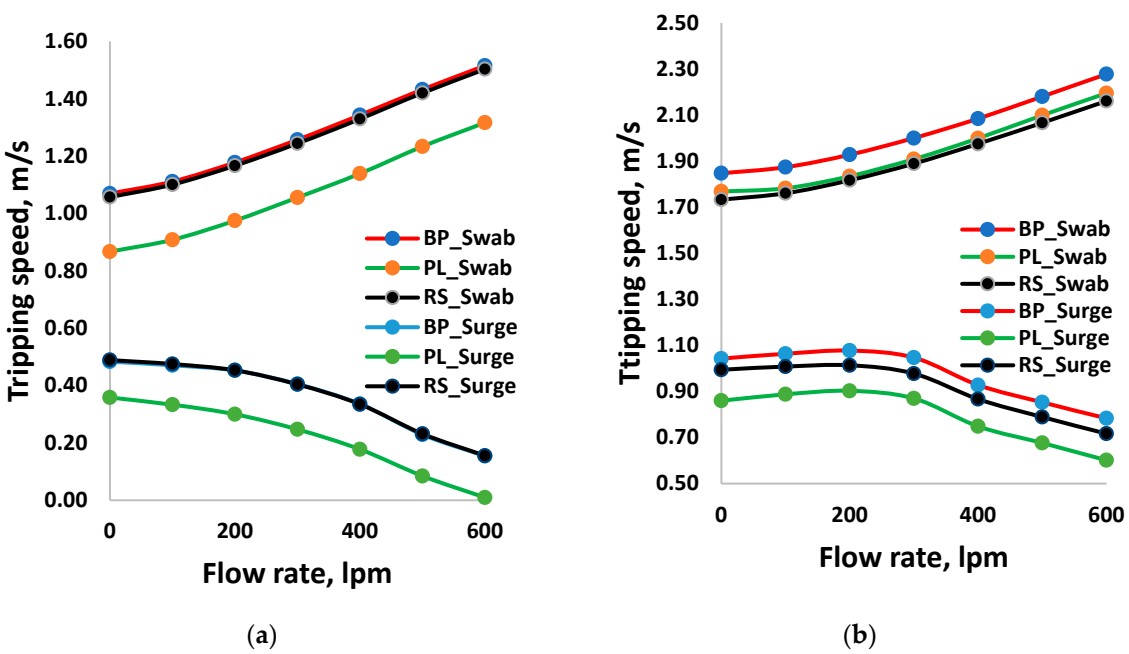

(**a**)                                                    (**b**)

**Figure 8.** (**a**) Example of swabbing and surging effects in the 2.0 sg 80:20 OBM filled in a deviated well. (**b**) Example of swabbing and surging effects in the 2.0 sg 90:10 OBM filled in adeviated well.

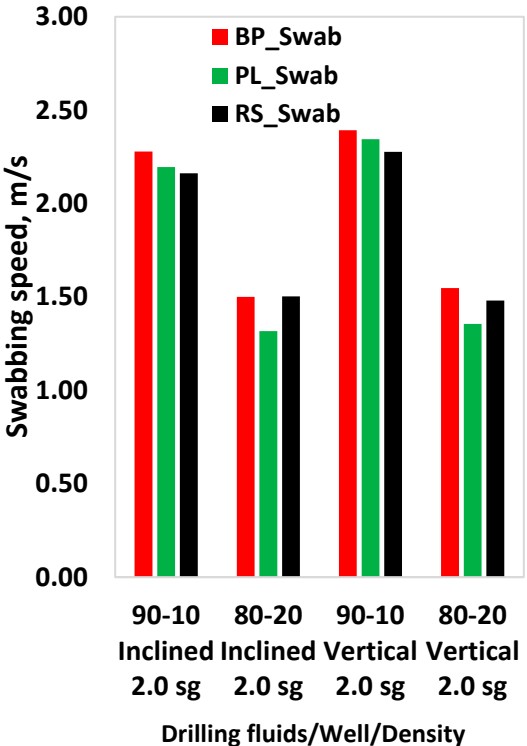

**Figure 9.** Swabbing comparisons of 2.0 sg 80:20 and 90:10 OBMs filled in deviated and vertical wells.

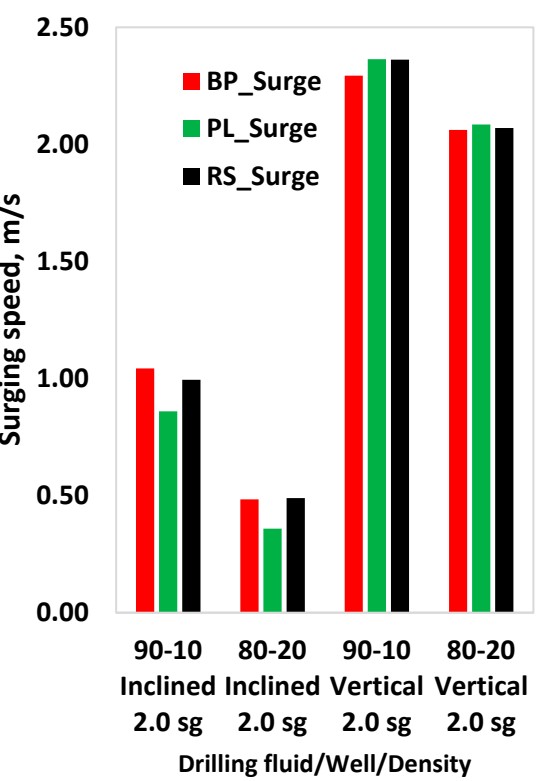

**Figure 10.** Surging comparisons of 2.0 sg 80:20 and 90:10 OBMs filled in deviated and vertical wells.

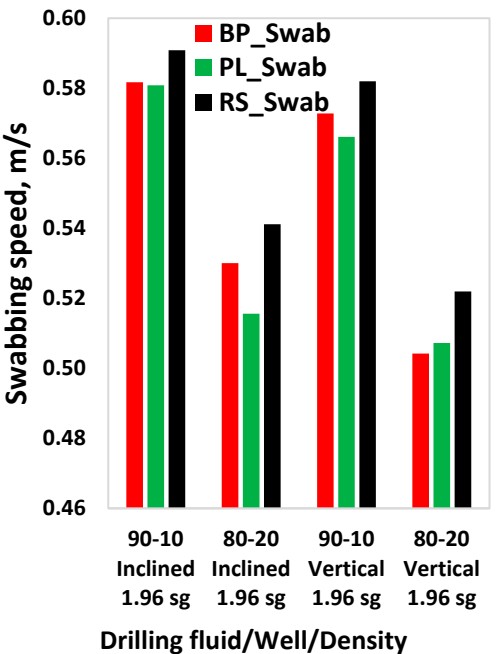

**Figure 11.** Swabbing comparisons of 1.96 sg 80:20 and 90:10 OBMs filled in deviated and inclined wells.

The percentile deviations of the BP from the PL and RS model predictions were calculated for better quantification. The comparisons are presented in Tables 5 and 6. As shown in Table 5, in vertical and deviated wells filled with the 2.0 sg 80:20 OBM drilling fluid, the BP swabbing speed model predictions were approximately 13.1% and 12.4% higher than the PL model, respectively. On the other hand, for the 2.0 sg 90:10 OBM in the inclined and vertical wells, the BP model predictions were approximately 5.1% and 4.8% higher than the RS, respectively. In the inclined well filled with 2 sg 90:10 and 80:20 OBMs, the surging speed model predictions deviated from the PL by 17.6% and 25.9%,

respectively. Table 6 shows the results obtained from the 1.96 sg drilling fluids. As shown, for both fluid types and wells, the BP deviation from the PL and RS recorded lower values.

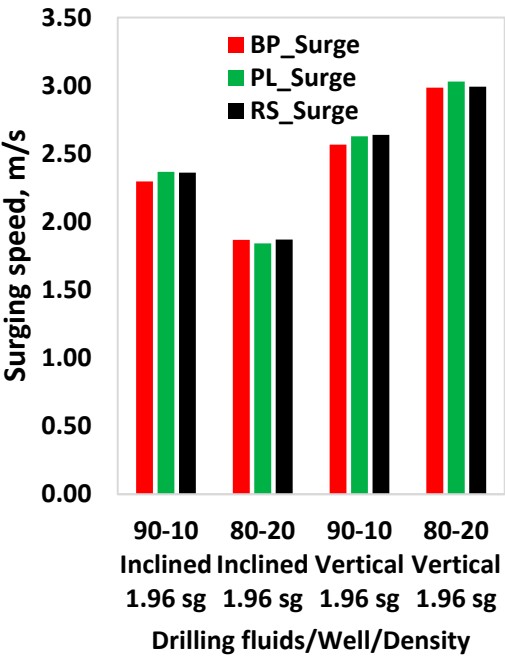

**Figure 12.** Surging comparisons of 1.96 sg 80:20 and 90:10 OBMs filled in deviated and vertical wells.

**Table 5.** Comparisons of 80:20 and 90:10 OBMs (from Figures 9 and 10, respectively) with 2.0 sg density in vertical and inclined wells.

| OBM/Well/Density | Swabbing | | Surging | |
|---|---|---|---|---|
| | % BP to PL Change | % BP to RS Change | % BP to PL Change | % BP to RS Change |
| 90:10 Inclined 2.0 sg | 3.63 | 5.12 | 17.58 | 4.66 |
| 80:20 Inclined 2.0 sg | 13.11 | 0.82 | 25.86 | −1.15 |
| 90:10 Vertical 2.0 sg | 1.97 | 4.82 | −3.06 | −2.97 |
| 80:20 Vertical 2.0 sg | 12.39 | 0.72 | −1.15 | −0.40 |

**Table 6.** Comparisons of 80:20 and 90:10 OBMs (from Figures 11 and 12, respectively) with 1.96 sg density in vertical and inclined wells.

| OBM/Well/Density | Swabbing | | Surging | |
|---|---|---|---|---|
| | % BP to PL Change | % BP to RS Change | % BP to PL Change | % BP to RS Change |
| 90:10 Inclined 1.96 sg | 0.14 | −1.58 | −3.09 | −2.84 |
| 80:20 Inclined 1.96 sg | 2.73 | −2.10 | 1.34 | −0.15 |
| 90:10 Vertical 1.96 sg | 1.16 | −1.60 | −2.36 | −2.80 |
| 80:20 Vertical 1.96 sg | −0.61 | −3.53 | −1.49 | −0.23 |

### 3.1.2. Result 2

The second computer simulation results were based on the drilling fluids shown in Figure 4. The examples displayed in Figure 13a,b are for the 80:20 and 90:10 OBMs filled in a deviated well, respectively. The 80:20 OBM results show that BP and RS swabbing and surging speed predictions were nearly equal and differed from the power law model for both fluid systems. However, deviations of the power law for swabbing were not consistent. Another observation was that for the 90:10 OBM, the surging speed for the flow

rate increments up to 300–400 lpm exhibited unexpected increases. For analysis purposes, the swabbing speeds (at 600 lpm) and surging speeds (no circulation) were considered; the results are displayed in Figures 14–17.

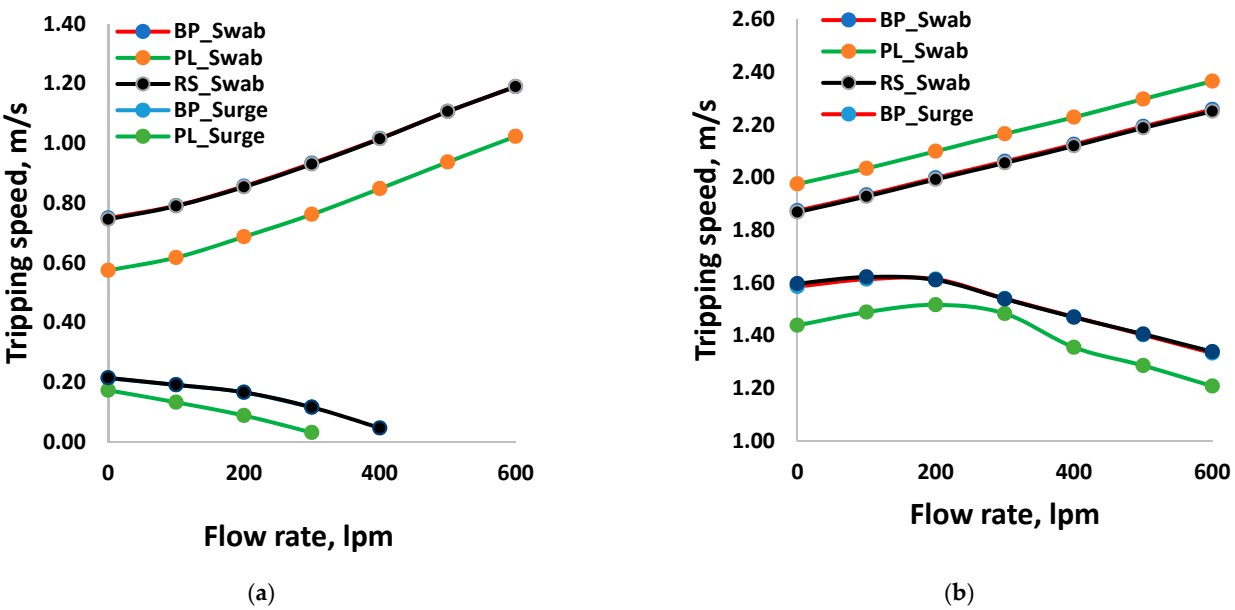

(**a**)  (**b**)

**Figure 13.** (**a**) Example of swabbing and surging effects of 2.0 sg 80:20 OBM filled in a deviated well. (**b**) Example of swabbing and surging effects of 2.0 sg 90:10 OBM filled in a deviated well.

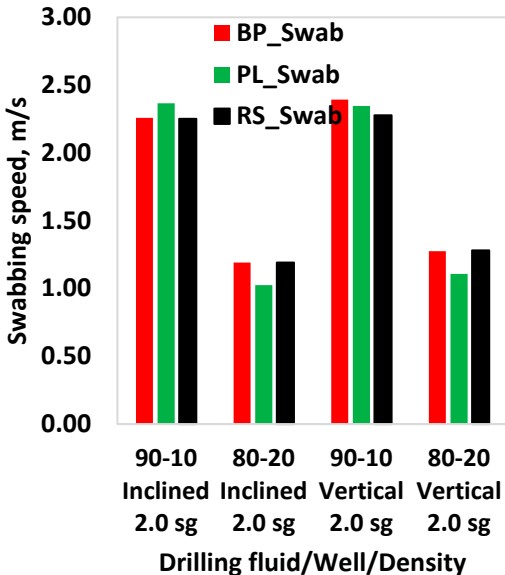

**Figure 14.** Swabbing comparisons of 2.0 sg 80:20 and 90:10 OBMs filled in deviated and vertical wells.

Similarly, the percentile deviations of the swabbing and surging speed predicted by the PL and RS models (i.e., Figures 14–17) were compared with the BP model. The comparisons are provided in Tables 7 and 8. In the deviated well filled with 2.0 sg 80:20 and 90:10 OBM, the results showed that the swab and surge speed predictions of the BP and RS is quite similar, recording less than 0.7% deviation. However, in the well filled with 80:20 OBM, the BP model's swab and surge percentile deviations from PL records were 14% and 18.8%, respectively. Similarly, the results obtained from the 90:10 OBM fluid showed −4.8% and 9.3% deviations, respectively. Table 8 also shows the results obtained from the 1.96 sg drilling fluids (80:20 and 90:10 OBM) filled in vertical and inclined wells. The results also show that the deviation of the BP from the RS was relatively lower than from the PL model.

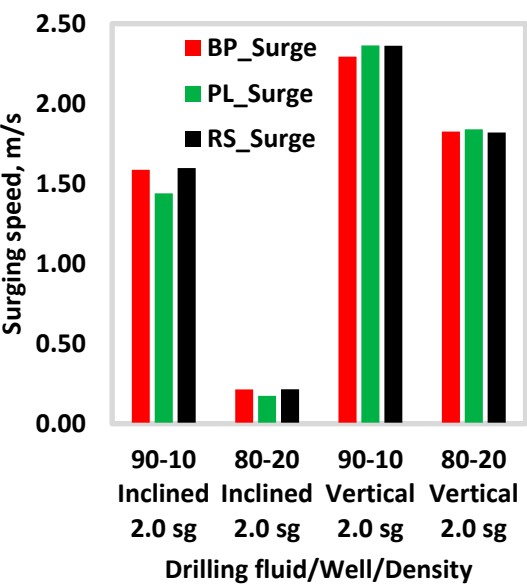

**Figure 15.** Surging comparisons of 2.0 sg 80:20 and 90:10 OBMs filled in deviated and vertical wells.

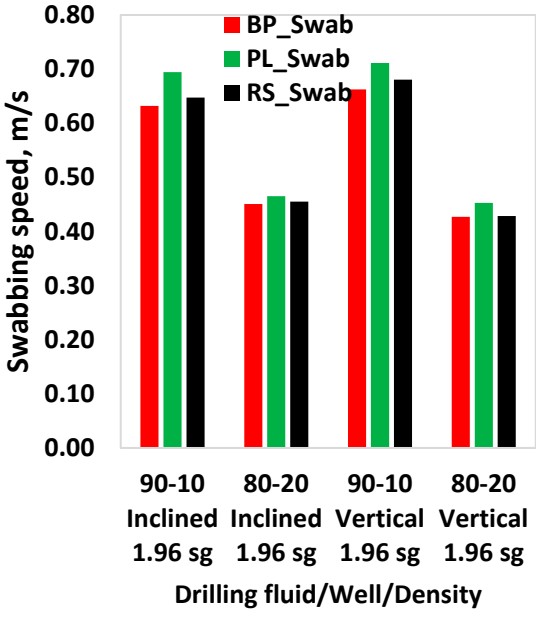

**Figure 16.** Swabbing comparisons of 2.0 sg 80:20 and 90:10 OBMs filled in deviated and vertical wells.

**Table 7.** Comparisons of 80:20 and 90:10 OBMs (from Figures 14 and 15, respectively) with 2.0 sg density in vertical and inclined wells.

| OBM/Well/Density | Swabbing | | Surging | |
|---|---|---|---|---|
| | % BP to PL Change | % BP to RS Change | % BP to PL Change | % BP to RS Change |
| 90:10 Inclined 2.0 sg | −4.77 | 0.30 | 9.28 | −0.70 |
| 80:20 Inclined 2.0 sg | 14.00 | 0.00 | 18.83 | −0.65 |
| 90:10 Vertical 2.0 sg | 1.97 | 4.82 | −3.06 | −2.97 |
| 80:20 Vertical 2.0 sg | 13.12 | −0.50 | −0.76 | 0.30 |

**Table 8.** Comparisons of 80:20 and 90:10 OBMs (from Figures 16 and 17, respectively) with 1.96 sg density in vertical and inclined wells.

| OBM/Well/Density | Swabbing | | Surging | |
|---|---|---|---|---|
| | % BP to PL Change | % BP to RS Change | % BP to PL Change | % BP to RS Change |
| 90:10 Inclined 1.96 sg | −9.89 | −2.42 | −4.55 | 0.14 |
| 80:20 Inclined 1.96 sg | −3.21 | −0.99 | 0.68 | −0.51 |
| 90:10 Vertical 1.96 sg | −7.34 | −2.73 | −2.96 | 0.05 |
| 80:20 Vertical 1.96 sg | −6.05 | −0.33 | −1.24 | −0.79 |

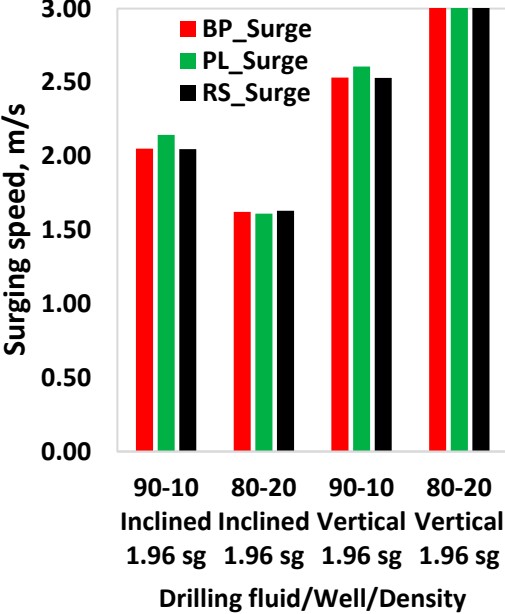

**Figure 17.** Surging comparisons of 2.0 sg 80:20 and 90:10 OBMs filled in deviated and vertical wells.

### 3.2. Machine-Learning-Based Modeling Result

#### 3.2.1. Result 1—Simulated and Laboratory-Based Data Model

The initial ML modeling was performed using a physics-based simulated dataset. The example presented here illustrates how ML models predicted software-generated dataset. For this, the drilling fluid shown in Figure 4 was circulated in the experimental well.

Figure 18 shows the simulated synthetic dataset obtained from the power law (PL) and Robertson–Stiff (RS) models. Simulation stopped at the maximum allowable tripping speed that caused the well pressure to reach the weak point (i.e., fracture at the casing shoe, including safety factor). Figure 19 also shows the laboratory scale surge pressure gradient as a function of the tripping speed [7].

Frictional pressure loss is proportional to the square of fluid flow. Similarly, the simulated (Figure 18) and the laboratory experimental (Figure 19) surge pressure variations with the pipe movement trend represented a second-order polynomial function. Therefore, the simulated and the experimental test data were modeled as a polynomial function that included both constant and middle terms. The maximum allowable well pressure was when the well pressure reached the casing pressure ($P_c$). The variation in well pressure due to tripping speeds, obtained from Equation (11), is given as follows:

$$P_c = \beta_0 + \beta_1 V_P + \beta_2 V_P^2 \tag{15}$$

where $\beta_0$, $\beta_1$, and $\beta_2$ are the curve fitting parameters, determined from the green or blue datasets.

Based on the knowledge of case shoe pressure, one can therefore write well pressure at the critical tripping speed using the inversion of Equation (15), given as:

$$V_P = \frac{-\beta_1 + \sqrt{\beta_1^2 - 4\beta_2(\beta_2 - P_c)}}{2\beta_2} \tag{16}$$

Using the synthetic and experimental data, polynomial-based modeling was performed to estimate the curve fitting parameters ($\beta_0$, $\beta_1$, and $\beta_2$); the results are presented in Table 9. Both models showed higher $R^2$ correlation values. However, it is essential to note that polynomial-based modeling was only applied for swab and surge field data or laboratory data if the pressure variations behaved as polynomials when the tripping speeds varied.

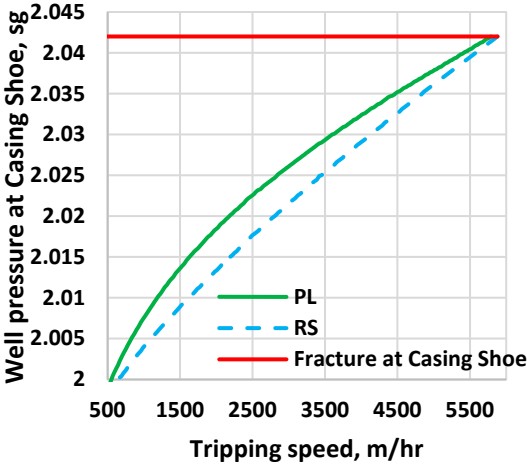

**Figure 18.** Drillbench-software-simulated surging pressure as a function of tripping speed up to the fracture point.

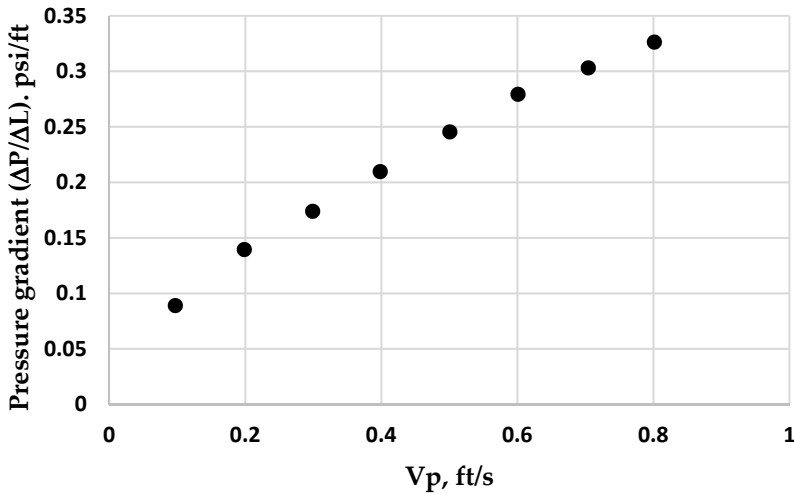

**Figure 19.** Experimental surge pressure gradient vs. tripping speed data [7].

**Table 9.** Polynomial function correlation factors of simulated and experimental data.

| Data | $\beta_2$ | $\beta_1$ | $\beta_0$ | $R^2$ |
|---|---|---|---|---|
| Figure 19 [7] | $-0.1675$ | $0.4851$ | $0.0452$ | $0.9992$ |
| Figure 18 [PL] | $-9 \times 10^{-10}$ | $1 \times 10^{-5}$ | $1.9954$ | $0.9965$ |
| Figure 18 [RS] | $-4 \times 10^{-10}$ | $1 \times 10^{-5}$ | $1.9941$ | $0.9997$ |

Subsequently, two machine learning algorithms (ANN and RF) were arbitrarily selected to evaluate how data-driven-based modeling predicted the synthetic, simulated data presented in Figure 18.

Figure 20 compares the simulated dataset and ANN model predictions. Table 10 summarizes the ANN model performance analysis. As shown in Table 10, the ANN model strongly correlated with the training and testing data, with $R^2$ values of 0.999 and 0.999, respectively. Moreover, the other statistical parameter also shows that the model perfectly predicted the synthetic dataset. Physics models generate data without including noise; the ML model prediction demonstrates the trustworthiness of the method.

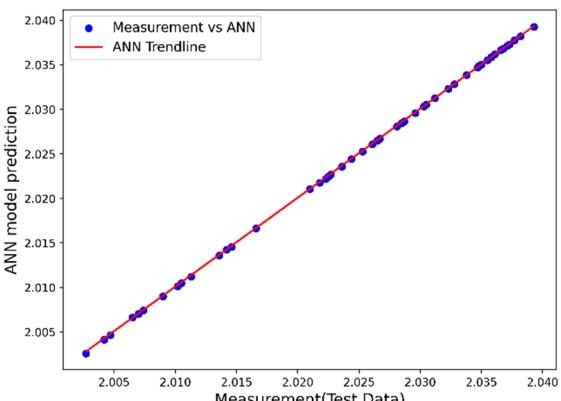

**Figure 20.** Scatter plot of 30% test data vs. ANN model prediction.

**Table 10.** Summary of the performance accuracy of ANN and random forest models.

| ML Models | Dataset | Model Performance Accuracy | |
|---|---|---|---|
| | | MSE | $R^2$ |
| ANN | Training | $2.63 \times 10^{-9}$ | 0.999 |
| | Testing | $2.35 \times 10^{-9}$ | 0.999 |
| RF | Training | $2.58 \times 10^{-8}$ | 0.999 |
| | Testing | $5.84 \times 10^{-9}$ | 0.999 |

Similarly, Figure 21 shows the results obtained from the simulated dataset and the RF model prediction. Table 10 also provides the RF model accuracy performance analysis. The $R^2$ values of the training and testing datasets showed strong correlations: 0.999 and 0.999, respectively. The MSE values indicate that the RF method accurately predicted the simulated data.

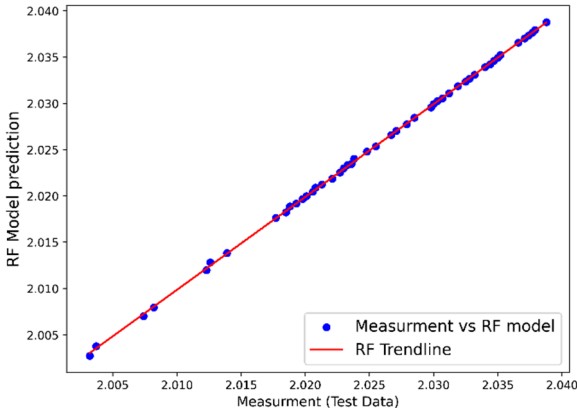

**Figure 21.** Scatter plot of 30% test data vs. RF model prediction.

### 3.2.2. Result 2—Field-Data-Based Machine Learning

In Section 3.2.1, the selected RF and ANN methods were shown to have significantly recovered the synthetic physics-based dataset. This section presents the results of the application of six ML models tested on field-measured data, as shown in Figure 5. The field data received were large, and in .txt format. The data were converted to Excel format and imported through the Pandas/Python library to perform appropriate data pre-processing. Initially, information about the dataset was explored to evaluate datatypes (object or float) and check whether an unrecorded dataset existed, as well as the presence of outliers. Data cleaning was then performed. Cross-correlation of the dataset was also evaluated to select the input features to be modeled with the target dataset.

The ML-based prediction model and the actual measured values were used for analysis of the statistical accuracy. For this, a linear regression model was derived, based on the model-predicted and actual values. Figures 22–27 show the ML model prediction and ML model linear trendline, given as:

$$\text{ML Model} = a.\text{Measured data} + b$$

where a and b are the slope and the intercept, respectively.

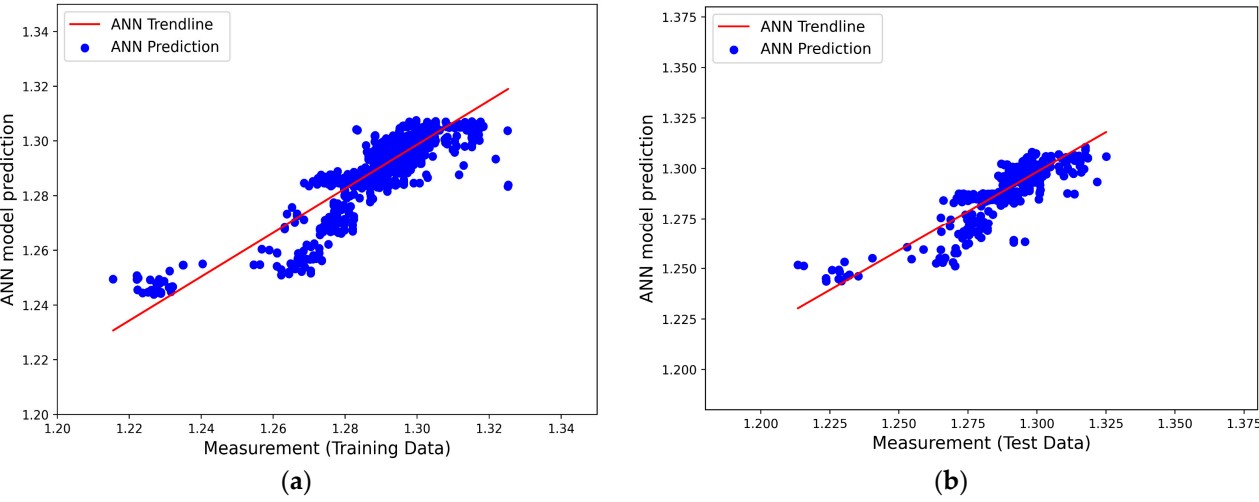

**Figure 22.** (**a**) Scatter plot of 70% training tripping-out data vs. ANN model prediction. (**b**) Scatter plot of 30% test tripping-out data vs. ANN model prediction.

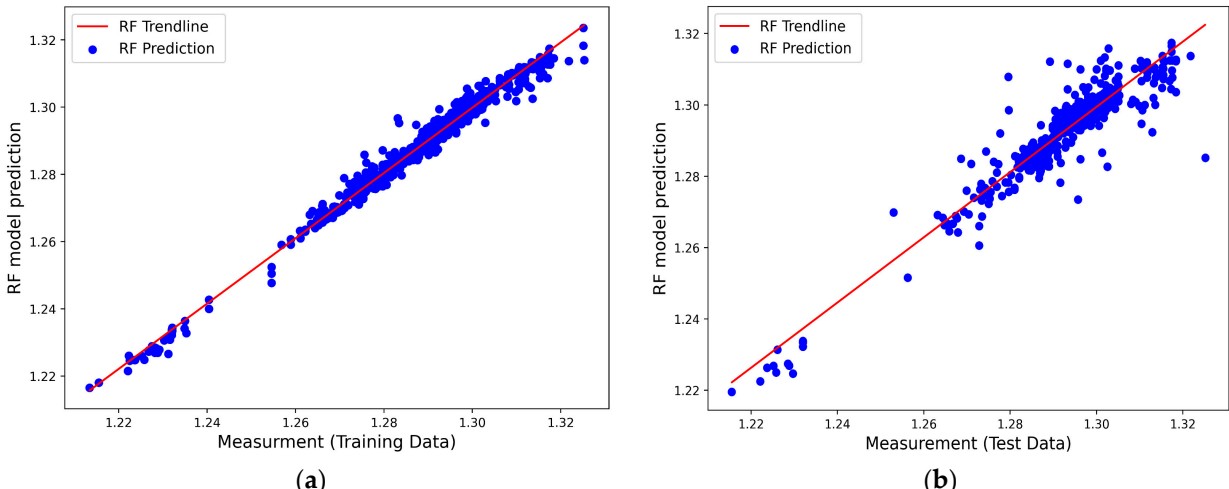

**Figure 23.** (**a**) Scatter plot of 70% training tripping-out data vs. RF model prediction. (**b**) Scatter plot of 30% test tripping-out data vs. RF model prediction.

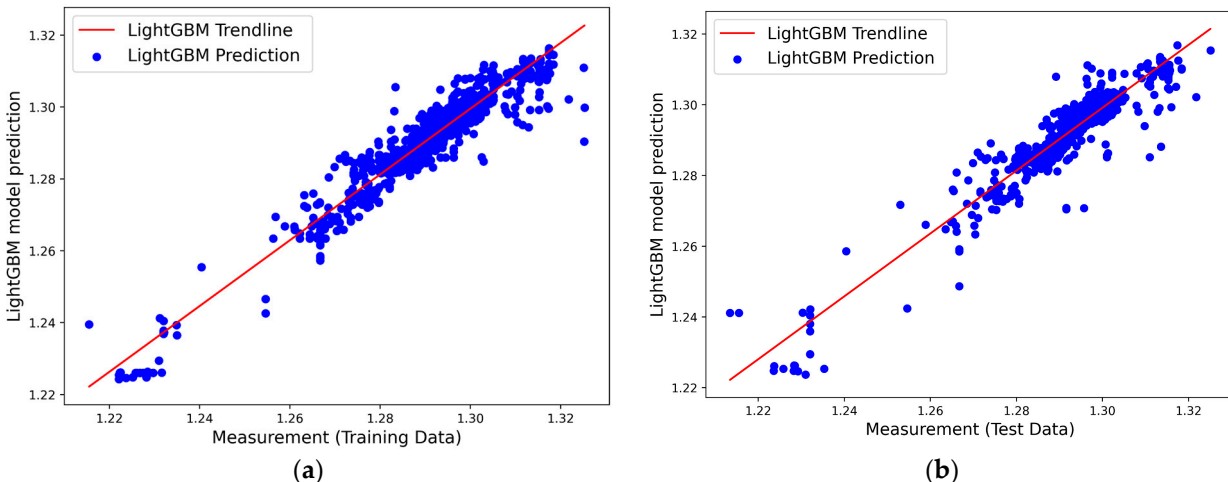

**Figure 24.** (**a**) Scatter plot of 70% training tripping-out data vs. LightGBM model prediction. (**b**) Scatter plot of 30% test tripping-out data vs. LightGBM model prediction.

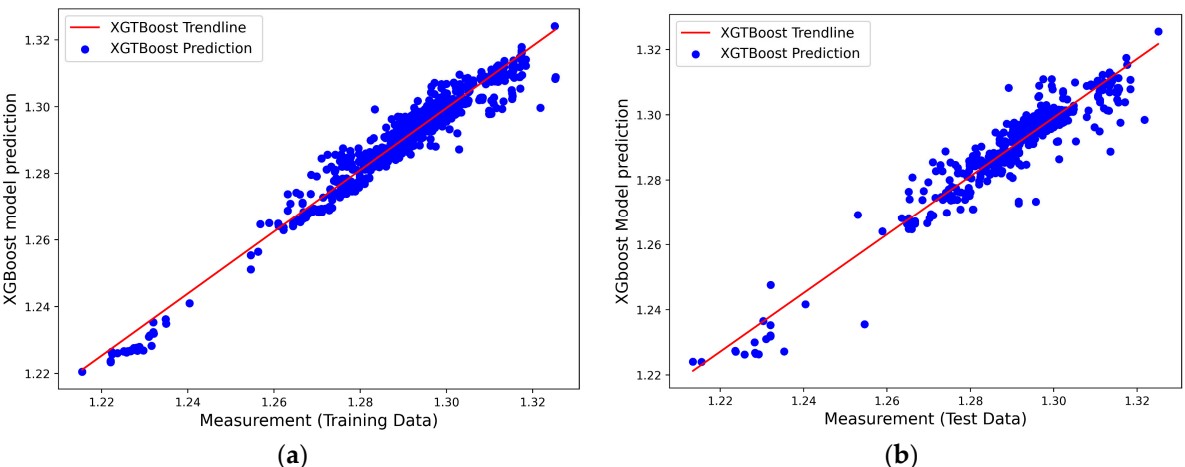

**Figure 25.** (**a**) Scatter plot of 70% training tripping-out data vs. XGBoost model prediction. (**b**) Scatter plot of 30% test tripping-out data vs. XGBoost model prediction.

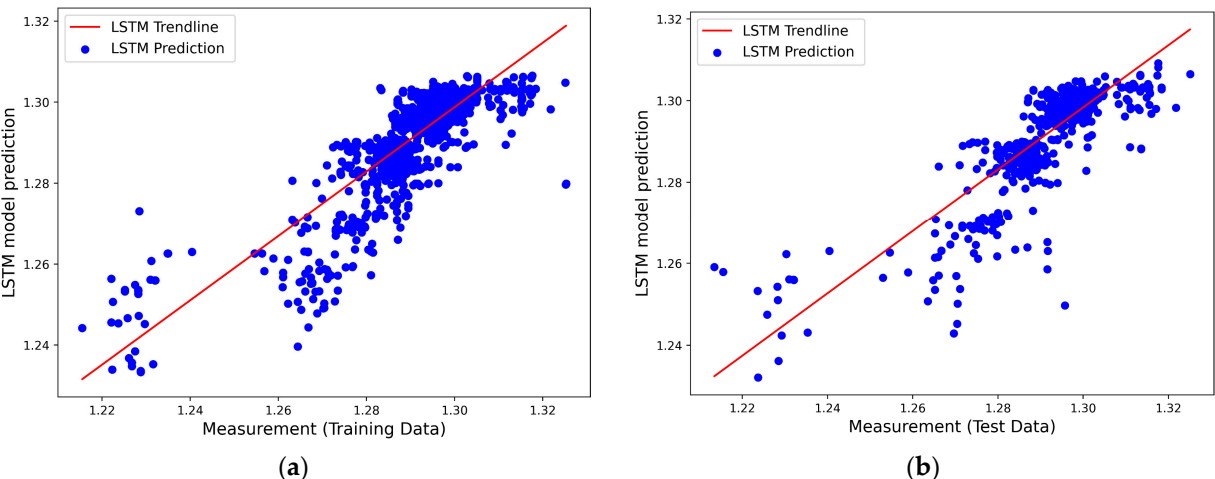

**Figure 26.** (**a**) Scatter plot of 70% training tripping-out data vs. LSTM model prediction. (**b**) Scatter plot of 30% test tripping-out data vs. LSTM model prediction.

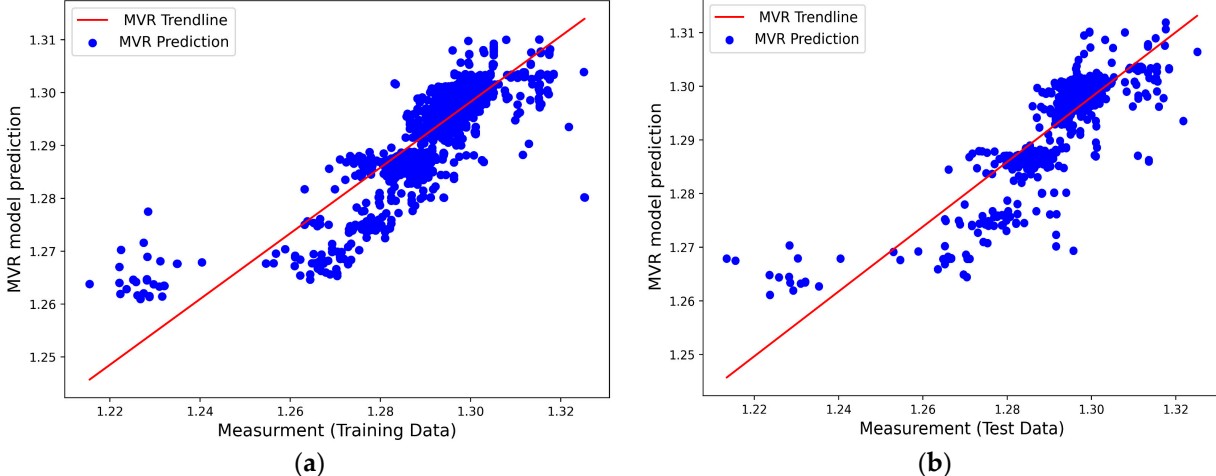

(a)

(b)

**Figure 27.** (**a**) Scatter plot of 70% training tripping-out data vs. multivariable model prediction. (**b**) Scatter plot of 30% test tripping-out data vs. multivariable model prediction.

Table 11 shows the ML model's accuracy performance with the training and test results displayed in Figures 22–27. The model predictions exhibited a high $R^2$ score and a minimum mean sum square error. The results demonstrate the potential application of machine learning modeling for the swab and surge field dataset.

**Table 11.** Performance accuracy analysis of the ML models with the tripping-out dataset.

| ML Model Algorithms | Dataset | Model Performance Accuracy | |
|---|---|---|---|
| | | MSE | $R^2$ |
| ANN | Training | $2.94 \times 10^{-5}$ | 0.7921 |
| | Testing | $3.43 \times 10^{-5}$ | 0.7836 |
| RF | Training | $1.95 \times 10^{-6}$ | 0.9879 |
| | Testing | $1.45 \times 10^{-5}$ | 0.8921 |
| LightGBM | Training | $1.05 \times 10^{-5}$ | 0.9256 |
| | Testing | $1.88 \times 10^{-5}$ | 0.8884 |
| XGBoost | Training | $7.01 \times 10^{-6}$ | 0.9504 |
| | Testing | $1.52 \times 10^{-5}$ | 0.9098 |
| LSTM | Training | $3.60 \times 10^{-5}$ | 0.7449 |
| | Testing | $4.56 \times 10^{-5}$ | 0.7299 |
| Multivariable | Training | $3.87 \times 10^{-5}$ | 0.7264 |
| | Testing | $4.86 \times 10^{-5}$ | 0.7120 |

## 4. Discussion

Tripping operations refer to lowering or withdrawing a drill string into a wellbore hole. The literature indicates that approximately 18% of drilling time is spent performing tripping operations, and is considered non-productive according to Christopher Jeffery et al. (2020) [29]. Tripping is performed, for example, to replace worn-out drill bits. Tripping string at a lower speed is safer concerning wellbore instability; however, it will increase the non-productive time and cost. Tripping at a higher speed will minimize the non-productive time, but pulling the strings above their optimum values will create undesired surging and swabbing pressures that could lead to wellbore fracture and collapse/kick, respectively. For safe and efficient operation, it is imperative to predict the appropriate swab surge pressure precisely associated with the optimized tripping speeds.

Amir et al. (2022) [39] presented an extensive literature review on surge/swab pressure models, which were developed based on physical laws (analytical models) and experimental works (empirical models). The developed analytical models were based on several

presumptions, and simulated in laboratory-controlled experimental setup conditions. However, it is difficult to precisely quantify the degree of drill string eccentricity, wellbore roughness, sizes, and fluid properties in actual drilling well operation conditions. Hence, model predictions with uncertain inputs will not be correct. Moreover, the model predictions also varied. This indicates the uncertainty of the parametric-based modeling for swab and surge models.

In this study, first, we conducted a physics-based swab and surge simulation to evaluate the model prediction in different well trajectories filled with drilling fluids exhibiting different rheological and physical properties. The models tested were Bingham plastic, power law, and Robertson–Stiff. Four oil-based drilling fluids with various viscosities were considered for the simulations, under both vertical and deviated well profiles.

Observations based on the considered drilling fluids and simulation setup revealed that:

- The model's predictions were inconsistent compared with each other.
- As the flow rates increased to approximately 200–300 lpm, surging speeds in the deviated well filled with 90:10 OBM showed an increasing trend, whereas in 80:20 OBM, the surging speed showed a decreasing trend. On the other hand, in both fluid systems, the surging speeds decreased when the flow rate increased above 300 lpm. Even though the trends in surge speeds for the three models' predictions seemed similar, the values were quite different.
- Regarding rheology fluid descriptions, the RS model showed a lower error deviation. However, the swab and surge percentile deviation from the BP was lower than the RS with PL. The swab and surge predictions with the three models varied in the different well trajectories filled with fluids of different densities and viscosities.
- It was difficult to conclude the accuracy of the hydraulics model prediction based on how the model accurately described the fluid rheological properties.

A general model that considers all physical processes and fluid properties is presently unavailable. It is, therefore, vital to calibrate models with a real-time measured dataset that considers bulk effects in the wellbore.

Hydraulics data measured in the North Sea oil field were compared with a hydraulic model by Lohne et al. in 2008 [40]. The comparison results revealed a difference between the measured and modeled values. This comparison demonstrated that the model was unable to predict the measurement. The model's parameters, including density, friction factor, and well path, were all ambiguous. Lohne et al. incorporated a calibration factor and set the friction factor value to only one because the model did not fully represent the physics. The authors created a dynamic calibration factor based on the collected data to calibrate the annulus and drill string pressure. Jeyhun et al. (2016) [27] deployed five hydraulic models on both laboratory and field data in a separate study; they discovered that the predictions derived from these models were inconsistent. For instance, the Herschel–Bulkley model may correctly forecast the hydraulics of fluid A in a pipe or annulus, whereas for fluid type B, Robertson–Stiff models could work better than the others. This suggests that no universal solution can anticipate a fluid's hydraulics in a wellbore due to the variable fluid flow and pipe movements in and out of the well.

For calibration of the hydraulics model, the accuracy of the measurement was also a critical factor. For this, Lohne et al. (2008) [40], Reeves et al. (2006) [28], and Christopher et al. (2020) [29] identified that the high-speed telemetry system, WDP, plays a significant role in terms of providing a higher rate of data transmission with less noise. Moreover, unlike measurement while drilling (MWD) sensors, Michael (2003) demonstrated that WDP telemetry performs continuous measurements of downhole pressure with low or zero drilling fluid flow rate conditions, [41]. The argument for data-driven-based modeling is that the measured data include all possible factors contributing to swab and surge impacts. Wired pipe technology enables high-quality and rapid data transfer. Moreover, knowing which physics-based modeling perfectly predicts the swab surge model as consistently and accurately as possible is difficult.

Therefore, the applicability of the selected ML models (ANN and RF) was first employed on the physics-based generated synthetic data. The results show that the ML perfectly correlated with the dataset.

The performance of the six ML algorithms implemented on unfiltered WPD tripping-out field datasets showed relatively accurate predictions on both the training and test datasets. However, after applying an exponential smoothing filter on the same field data, the models demonstrated excellent performance.

## 5. Conclusions

Optimized swab and surge operations contribute to a reduction in tripping time, and hence, reduce non-productive time. This paper evaluates physics-based swab and surge simulation and the data-driven-based machine learning modeling of field surging field datasets.

The summary of the analysis shows that:

- Deviations of swab surge model predictions from each model are inconsistent.
- Physics-based models generally require model calibration based on accurately measured data.
- The reviewed physics models do not consider all operational parameters, constraints, fluid properties, and non-uniform eccentricity. Moreover, it is difficult to quantify these parameters precisely in a drilling well.
- Data-driven-based modeling predicts both training and unseen test data with higher accuracy.

The application of ML models on the swabbing pressure example presented in this paper indicates the potential of deploying an intelligent solution as online- or offline-based modeling. This will enable drilling engineers to perform drilling optimization and better plan wells.

**Author Contributions:** A.M. Conceptualization, Methodology, Data processing, Software simulation, ML modeling, Testing, Result Analysis, Interpretation, and Writing. M.B. Methodology, Draft Manuscript Preparation, Supervision, Review, and Editing. R.D. Supervision, Project administration, Funding acquisition, Review, and Editing. All authors have read and agreed to the published version of the manuscript.

**Funding:** No external funding.

**Institutional Review Board Statement:** Not applicable.

**Informed Consent Statement:** Not applicable.

**Data Availability Statement:** Not applicable.

**Acknowledgments:** We thank Schlumberger for providing free Drillbench software to conduct the swab/surge simulation.

**Conflicts of Interest:** The authors declare no conflict of interest.

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
