# Peer review of "Physics-Based Swab and Surge Simulations and the Machine Learning Modeling of Field Telemetry Swab Datasets"

_applsci, doi:10.3390/app131810252_

Round 1

Reviewer 1 Report

Dear Authors,

Thank you for submitting your work! The work has quality, but in my opinion, more details about the data used in the work, as well as more detailed explanations of why some criteria were chosen, are missing.

 Comments and recommendations for correction of the work would be:

Comment 1:

Line 29- MWD?

Line 38- BHA

Can you write the full meaning of the abbreviation? The first time it appears in the text you should explain the full meaning.

Comment 2:

Line 154- I think it would be more correct to write it: “The method was initially developed by Chen et al. (2016)” instead “The method was initially developed by (Chen et al. 2016)”.

Comment 3:

Line 166- It would be better if you wrote “It has a great capacity to capture the underlying correlations among the different features in the data [25]“ instead „It has a great capacity to capture the underlying correlations among the different features in the data (Chujie Tian et al. 2018) [25].“

Comment 4:

Line 168- “implementation. [26] We”

Is reference 26 valid for the previous or next sentence? Number [26] is written after the full stop, so that's why I'm asking.

Comment 5:

Line 233-„2.06 sg“ What does sg stand for? You must write when it appears for the first time in the text.

Comment 6:

Align image names to be unique. Some images have a space between the word Figure__ and the name, while some do not.

Figure 4:Pore Pressure and Formation Strength Prognosis

Figure 5:Experimental well structure and well trajectory

Comment 7:

Write and explain the difference between figures 6 and 7 because they have the same picture name.

Comment 8:

Table 5 and Table 6- Please standardize the labeling of the tables (first of all, the font and then the spaces in certain places).

Comment 9:

Table 7 and Table 8- Please standardize the labeling of the tables (first of all, the font and then the spaces in certain places).

Comment 10:

Line 544- “or blue dataset “ A period is missing at the end of the sentence.

Comment 11:

Line 693- The tripping operation is lowering or withdrawing the drill string into the hole. The literature indicates that about 18% of the time is spent in tripping operations and is considered non-productive Christopher Jeffery et al.

Missing “according to Christopher Jeffery et al.”

Comment 12:

In chapter 2.1.1. you listed 6 models and a few sentences before (Line 105) you listed 7 models (including Polynomial,).  Please explain that.

Comment 13:

Line 21- What do points DHT 003, DHT 002 and DHT 001 represent in Figure 2.? If these are the locations of the sensors, it seems to me that they do not correspond to the values written in the text (67 m, 328 m, 575 m).

Comment 14:

Figure 3 - Please explain all the diagrams shown in more detail through the text.

Comment 15:

Figure 5a- Is it possible to make the depth dimensions more visible? They seem too elongated like this.

Comment 16:

Does the type of soil have an effect on what the results will be? You have not displayed any data.

Comment 17:

Line 556- Why are they in chapter 3.2.1. (Result 1-Simulated and Laboratory-based data model) only ANN and RF models shown? Why were only those two models chosen? Based on what?

Comment 18:

Can you give more details about the data that was used other than that it was obtained from Norway and Bangladesh?

Author Response

Please see the attached "point-to-point" response. 

Reviewer 2 Report

1.      I find Figure 1 confusing. What do the three arrows in the Training data represent? What about the three arrows intersecting above Model Prediction? The author needs to revise Figure 1 for better clarity.

2.      In section 2.1.1, the author introduced some algorithms used in the study. However, there are certain algorithms such as Random Forest, XGBOOST, and LSTM that were not explained in detail. Please provide a thorough explanation of the roles of these algorithms in the context of this paper, as well as the training and testing details, instead of just referencing the work of others.

3.      Why does the phrase "LightGBM algorithm employed to the field dataset" appear in both line 150 and line 161? The author needs to carefully review the paper to avoid such occurrences. Additionally, it is recommended to avoid using fourth-level headings. The hierarchical structure of the headings in this article is confusing.

4.      In section 2.2, the author introduced four evaluation metrics used in the study. What are the differences between these metrics? What aspects of the model's performance do they each measure? Please provide additional explanations.

5.      Why is "Physics-based modeling" listed as section 2.1 after section 2.2?

6.      From figures 23, 25, 27, and 28, it can be observed that the fitting of ANN, LightGBM, LSTM, and Multivariate model prediction is poor. How did the author train these models? Please provide a detailed explanation of the model training process.

7.      I have some doubts regarding the author's statement that the dataset was divided into a 70% training set and a 30% testing set. Firstly, how did the author ensure that there is no data leakage, imbalance, or distributional differences during the dataset splitting process? Additionally, what is the size of the dataset used by the author? In what format does the data exist? Moreover, since the author employed the sequential model LSTM, how did they ensure that the data was split in chronological order to avoid future information leakage into the model during training? The most crucial point is whether the author separately allocated a validation set within the training set. Without a validation set, how can the fitting status of the data-driven model be assessed? Did the author optimize the model's hyperparameters using the testing set? Please provide a detailed explanation.

Moderate editing of English language required

Author Response

(The authors gave the same response as above.)

Reviewer 3 Report

The authors presented the manuscript with the title “Physics-Based Swab and Surge Simulations and Machine Learning Modelling of Field Telemetry Swab Dataset”. Most parts of the article appear to be correct.

Some notes on the manuscript:

1. The abbreviations used should have an explanation in the text when they are used for the first time.

2. Not all methods used have references to the literature.

3. On some charts, you need to add a standard deviation.

4. The manuscript is very extensive, some information should be added as supporting material.

5. The authors should consider combining the discussion with the results, the presented form makes it difficult to understand and the need to go back to selected fragments.

6. Extend the discussion by enriching it with literature references.

The manuscript may be published in the Applied Science after a major revision.

Author Response

(The authors gave the same response as above.)

Round 2

Reviewer 1 Report

Dear Authors,

Thank you for the submitted version of the work! I have reviewed your responses. What is a huge problem is that you have not matched the answers with the corrected version of the paper. In your answers, you emphasized that you implemented my recommendations, but with an insight into the work, you concluded that you did not do that.

Also, in the corrected part of the work, you made mistakes in terms of typos and duplication of punctuation marks.

The noted shortcomings are as follows:

 Comment 1:

You did not fully correct the work according to my comment 1. Line 29- MWD?

My comment in the previous version and your response:

Reviewer-1 Comment 13: Line 21- What do points DHT 003, DHT 002 and DHT 001 represent in Figure 2.? If these are the locations of the sensors, it seems to me that they do not correspond to the values written in the text (67 m, 328 m, 575 m).

 Authors’ reply: These are the sensor placements; we have correlated the sensors shown in the figure explaining in the test corresponding to the sensor placement.

 Comment 2:

You did not apply what you wrote in response to my comment 2.

My comment in the previous version and your response:

Reviewer-1 Comment 2: Line 154- I think it would be more correct to write it: “The method was initially developed by Chen et al. (2016)” instead “The method was initially developed by (Chen et al. 2016)”.

Authors’ reply: We have corrected it; Thank you!

 Comment 3:

You did not apply the correction in the way you wrote in your reply to my comment 3. Now it's line 369 and reference 37.

My comment in the previous version and your response:

Reviewer-1 Comment 4: Line 168- “implementation. [26] We”

Is reference 26 valid for the previous or next sentence? Number [26] is written after the full stop, so that's why I'm asking.

Authors’ reply: The citation should come before the full stop; we have corrected it; thank you!

 Comment 4:

You have applied the correction from comment 6 to the listed figures, but still Figure 1 remains.

My comment in the previous version and your response:

 Reviewer-1 Comment 6: Align image names to be unique. Some images have a space between the word Figure__ and the name, while some do not.

Figure 4:Pore Pressure and Formation Strength Prognosis

Figure 5:Experimental well structure and well trajectory

Authors’ reply: We agree with your comments. As per your comments, we have corrected it!

  Comment 5:

You did not correct according to the recommendation from comment 10 even though you wrote that you did.

 My comment in the previous version and your response:

Reviewer-1 Comment 10: Line 544- “or blue dataset “ A period is missing at the end of the sentence.

 Authors’ reply:  Well observed!  We appreciate it. As per your valuable comments, we have corrected it.

Comment 6:

You did not correct according to the recommendation from comment 11 even though you wrote that you did. (now line 734)

My comment in the previous version and your response:

Reviewer-1 Comment 11: Line 693- The tripping operation is lowering or withdrawing the drill string into the hole. The literature indicates that about 18% of the time is spent in tripping operations and is considered non-productive Christopher Jeffery et al.

Missing “according to Christopher Jeffery et al.”

Authors’ reply:  Well observed!  We appreciate it. As per your valuable comments, we have corrected it.

Comment 7:

Figure 3 & 4 -Is it possible to change the name of the figures so that they are not the same?

Comment 8:

Line 291- The sentence has errors (double comma, double period). “Finally, the model performance accuracy was also evaluated  with a coefficient of determination (R2), , and mean square error. . All”

Comment 9:

Line 293- “es.Th”   

Missing space between sentences. Recheck the whole text because I noticed similar mistakes in some other places.

Comment  10:

Line 298- Figure 5 name- modeling instead moeling

Comment 11:

?0  -  in the explanations of the coefficients, the same coefficients and signs are not written as in equation 12.

Comment  12:

Written explanation for comment 16 from review 1 I did not find that you added in the text and you wrote that you added.

Author Response

Dear Reviewer, thank you very much for the valuable comment on improving the quality of our manuscript!

Please find author reply in attachments.

Reviewer 2 Report

The author provided detailed explanations to most of my inquiries. However, regarding the crucial question of whether a validation set was used (#6f), the author did not provide a clear explanation. While it is true that the Adam optimizer generally has lower dependence on hyperparameters compared to SGD, it does not guarantee perfect model fitting. In fact, using Adam is more likely to result in overfitting, especially for small or sparse datasets. Therefore, during the training process, it is necessary to further partition the training data into a validation set. This step is commonly practiced in various research fields involving artificial intelligence models. The validation set assists in hyperparameter tuning, prevents overfitting, and monitors the training process. It is incorrect to directly adjust training hyperparameters using the test set. This would lead to data leakage issues and an inaccurate assessment of the model's true performance. If possible, the author should provide experimental logs, training curves, raw data, or other types of evidence to demonstrate the use of a validation set in the model training process. If a validation set was not used, an explanation should be given.

Minor editing of English language required

Author Response

(The authors gave the same response as above.)

Reviewer 3 Report

Manuscript can be published in present form in journal.

Author Response

(The authors gave the same response as above.)

Round 3

Reviewer 1 Report

Dear Authors,

Thank you for submitting the corrected version of the paper. The answers to all comments are satisfactory and, unlike last time, you applied everything in the corrected version of the paper and not only in the cover letter. This version is much more understandable with additional explanations. Also, it remains to review the text for technical errors. I noticed that certain words in the keywords are bolded, but this is negligible.

Author Response

We have through the manuscript and fixed some residual errors.  

Thank you, Sir or Madam, for contributing many valuable suggestions to make our paper more qualitative; And thank you, for accepting our paper. 

Amir Mohammad

(on behalf of the authors)

Reviewer 2 Report

The author has made tremendous efforts to enhance the quality of the manuscript, and I currently have no doubts. I recommend accepting it.

Author Response

Thank you, Sir or Madam, for contributing many valuable suggestions to make our paper more qualitative; And thank you, for accepting our paper. 

Amir Mohammad

(on behalf of the authors)